# Clinical Studies Regarding Electromagnetic Stimulation in Proximity of Dental Implants on Patients with/without Orthodontic Treatment

**DOI:** 10.3390/jcm9123983

**Published:** 2020-12-09

**Authors:** Eugen-Silviu Bud, Anamaria Bud, Mariana Păcurar, Alexandru Vlasa, Ana Petra Lazăr, Luminita Lazăr

**Affiliations:** Faculty of Dental Medicine, University of Medicine and Pharmacy, Science and Technology George Emil Palade Târgu-Mureș, 38 Gheorghe Marinescu Street, 540139 Târgu Mureș, Romania; eugen.bud@gmail.com (E.-S.B.); anamaria_jurca@yahoo.com (A.B.); marianapac@yahoo.com (M.P.); analazar_mvfa@yahoo.com (A.P.L.); luminita.lazar15@gmail.com (L.L.)

**Keywords:** dental implant, conventional healing caps, pulse electromagnetic healing caps, bone density

## Abstract

As a result of the loss of a tooth, there is a decrease in trabecular bone and loss of height and width of the adjacent bone. This study was designed as an observational imaging study, regarding structural changes that may occur during healing after the placement of Titanium dental implants. For this purpose, Cone Beam Computed Tomography was used in order to determine bone modifications around dental implants, loaded either with conventional healing caps or with healing caps pulsating electromagnetic waves, Magdent™, Haifa, Israel. The mean age of the study population was 49.84 ± 3.29 years (95% confidence interval (CI): 46.55−53.13). According to the voxel measurements after conventional treatment, there was a significant difference *p* < 0.0001 between bone radiodensity before treatment 288.1 ± 47.16 Standard Deviation (SD), and bone radiodensity 688.1 ± 81.02 SD after treatment with conventional healing caps. According to the voxel measurements after treatment with MagdentMed™ pulse electromagnetic healing caps, there was a significant difference *p* < 0.0001 between bone radiodensity before treatment 310.7 ± 53.26 SD and bone radiodensity after treatment with MED caps 734 ± 61.96 SD. The most common result of our study was a slightly higher radiodensity nearest the interface of dental implants after treatment.

## 1. Introduction

The human periodontium is a dynamic complex of tissues that undergoes remodeling in the area after the extraction of a tooth, because the alveolar bone requires mechanical stimulation to maintain its shape and density. Following the loss of a tooth, there is a decrease in trabecular bone and loss of height and width of the surrounding bone. Bone loss has been well documented by many researchers in human models [1,2,3,4] and in animal ones [5] and the results of these studies conclude that the remaining bone that undergoes these changes is difficult to be restored to the initial function. All these periodontal structures are dependent on each other; as a result of the extraction of a tooth, there is also a loss of gingival structure, leading to a reduction in soft tissue. As a result of a loss of a tooth there is also a decrease in space on the dental arches due to the movement of the neighboring teeth adjacent to the area of extraction. This movement is dependent on few factors such as: occlusal contacts, position on the arch of the lost tooth, age of the patient, time passed from the moment of extraction and dental-maxillary abnormalities. The decreasing space on the dental arches can be countered by space maintainers, prosthetic treatment or orthodontic treatment. Orthodontic treatment can be also used to open space on the dental arches if it is lost due to the physiological mesialization of the tooth or to other different types of movements like rotations or extrusions [6,7].

The quality of bone depends on the anatomical location, age of the patient, general health status, and local infections and tumors that may influence the quality of the bone. The most dense bone type is found in the anterior part of the mandible, followed by the posterior part of the mandible, the anterior part of the maxilla, and the posterior part of the maxilla. Assessment of bone density was difficult in the past, but an important advancement in bone quality estimation was made using computed tomography (CT), by characterizing bone density in terms of the CT number (or radiodensity), expressed in Hounsfield units (HU) [8]. The CT number describes the ability of a substance to attenuate an X-ray beam, ranging from –1000 HU for air to about 3000 HU for enamel. A significant correlation was found between the observed radiodensities of the implant sites and their subjective bone density scores [9]. Further studies [10] have established the ranges of HU values corresponding to each bone density class: D1 bone > 1250 HU, D2 bone 850–1250 HU, D3 bone 350–850 HU, D4 bone 150–350 HU and D5 bone < 150 HU.

This study was designed as an observational imaging study, regarding structural changes that may occur during healing after the placement of dental Titanium implants. For this, Cone Beam Computed Tomography was used in order to determine bone modifications around dental implants, loaded either with conventional healing caps or with healing caps pulsating electromagnetic waves, (Magdent™, Bnei-Brak, Israel). 

## 2. Materials and Methods

The study was planned as a split-mouth controlled study, on 29 patients in need of oral rehabilitation based on dental implants, using the following protocol:

The inclusion criteria for the treatment were:(1)Age between 30–60 years;(2)Kennedy class I, II or III edentation;(3)Healthy subjects.

The exclusion criteria were:(1)Poor oral hygiene;(2)Smoking;(3)The presence of systemic diseases that could affect the outcome of surgical treatment;(4)Subjects suffering from diabetes.

All procedures followed were in accordance with the ethical standards of the responsible committee on human experimentation (institutional and national) and with the Helsinki Declaration of 1975, as revised in 2008. Informed consent was obtained from all patients for inclusion in the study. The Ethics committee of Algocalm Private Medical Center of Târgu Mureș, Romania, approved the study (898/29.05.2020).

In the subsequent stage of the study, the clinical objectives were met, namely the determination of the radiological bone level, width, height and bone density on Cone Beam Computed Tomographyscans (Planmeca™, Helsinki, Finland). CBCT images were recorded using a tube voltage of 89 kV and a current intensity of 6 mA, the images were recorded with a cylindrical field of view (FOV) of 82 mm both in diameter and height (Figure 1), the voxel size was 0.2 × 0.2 × 0.2 mm and the dental arches were positioned similarly in the FOV, as presented in Figure 2.

For image acquisition, we used a ProMax 3D CBCT unit (Planmeca, Helsinki, Finland), with the previously mentioned settings. Images were acquired and saved in JPEG format. To locate implant target sites, bone density, and to record the mean CT number of the trabecular bone from these sites, we used the OnDemand 3Ddata App™ software (Cybermed Inc., Seoul, South Korea). In each site, our region of interest was a square volume of bone located within the alveolar ridge (Figure 3, Figure 4 and Figure 5). All data recorded were input into the Microsoft Office Excel™, 2017 version, analysis software (Microsoft, Redmond, WA, USA).

Each voxel of the CBCT volume is characterized by a CT number, expressed in HU (Hounsfield Units). The software displays the mean value of the CT numbers of the constituent voxels and the standard deviation of these CT numbers (Figure 5).

After recording the preliminary clinical situation, 8 patients from the study group (27.5%) underwent fixed orthodontic treatment for space opening in order to insert implants in the best possible position. Fixed orthodontic treatment varied from 8 weeks to 17 weeks and the medium space gained was approximatively 2.3 mm. In the subsequent stage of the study, the patients underwent initial periodontal treatment, including manual and ultrasonic debridement and cleaning, training on the correct brushing techniques, and the use of oral hygiene aids.

### 2.1. Surgical Stage

After re-evaluating the subjects 6 weeks post-sanitization and confirming the need for surgical therapy, all patients underwent the following surgical protocol:(1)Anesthesia of the concerned area by:Infiltration.Peripheral trunk anesthesia.(2)Incision: a horizontal incision on the alveolar ridge package ± 1–2 vertical incisions, starting from the horizontal one towards the bottom of the vestibular mucosa.(3)Detachment and reflection of the mucoperiosteal flap on the vestibular and/or oral face of the tooth; to maintain the viability of the flap, saline irrigations were performed.(4)Insertion of MIS Seven™ dental implants, MIS Dental Implants, Israel, standard platform, according to the bone characteristics for each individual site, based on the CBCT scans. After insertion, a resonance frequency analysis RAF was performed using an implant stability tester Osstell™ ISQ (Osstell™, Gothenburg, Sweden). Implants stability ranged from 58 ISQ to 74 ISQ values on the tester.(5)Loading of dental implants either with conventional healing caps, or with pulsating electromagnetic caps (Magdent™, Bnei-Brak, Israel) (Figure 6 and Figure 7).(6)Repositioning and suturing the flap.

Postoperative indications
(1)Patients were instructed to use an antiseptic mouthwash daily for 3 weeks (e.g., chlorhexidine solution 0.1–0.2%) and antibiotic therapy was instituted;(2)The sutures were removed when the postoperative wound was clinically healed and when they were no longer needed to keep the flap in place (7–10 days);(3)Patients were advised not to brush the area until 2 weeks postoperatively, professional brushing was performed if necessary; they were also recommended a diet that did not require dental units for hard chewing;(4)At 2 weeks, the patients were retrained regarding dental brushing techniques, including interproximal brushing;

### 2.2. Healing Stage

In the next phase of the study, immediately after implantation, healing caps were placed in position in order to achieve the best healing possible. The study group was divided into two sub-groups as follows:

Subgroup 1 consisted of 28 dental implants placed in all 12 patients included in the study, covered with conventional/standard healing caps (Figure 6), and subgroup 2 consisted of 25 dental implants covered with pulse induction electromagnetic healing caps (Magdent™, Bnei-Brak, Israel) (Figure 7 and Figure 8) placed in all patients included in the study. No differentiation between the patients was made, all patients received implants with conventional healing caps and pulse electromagnetic caps randomly. 

Magdent’s™ MED (MagdentMed) novel technology utilizes electromagnetic fields to stimulate, accelerate, and improve bone formation and quality for more successful dental implant procedures in shorter time. MED consists of a battery, an electronic device, and a coil that fits most implant models in much the same way as current simple/conventional healing abutments.

After 60 days of healing process, new CBCT images were recorded in order to assess bone density changes that might have occurred near the implant sites (Figure 9, Figure 10 and Figure 11).

In the next phase of the study, after a healing period of 60 days, new measurements regarding bone density were recorded (Figure 12 and Figure 13). To avoid the metal artifact/beam hardening effect created by implants, new measurements were recorded at a 2 m distance from the implant site.

## 3. Statistical Analysis 

Statistical processing was performed using GraphPad Prism™ V6.01 software for Windows™ (Microsoft, Redmond, WA, USA). Statistical analysis involved the use of the Student’s *t* test for unpaired (independent) and paired (dependent) data. The Kolmogorov–Smirnov test was used to test the normality of the data. The chosen significance threshold was alpha = 0.05, considering *p* significant when *p* < 0.05.

## 4. Results

Twenty-nine patients (15 males and 14 females) were enrolled in the study and all of them showed modifications regarding bone density around dental implants. The mean age of the study population was 49.84 ± 3.29 years (95% confidence interval (CI): 46.55–53.13). 

According to the voxel measurements after treatment with conventional caps (Figure 14), there was a significant difference *p* < 0.0001 between bone radiodensity before treatment 288.1 ± 47.16 Standard Deviation (SD) and bone radiodensity 688.1 ± 81.02 SD after treatment.

There were no significant differences *p* = 0.59 between the 2 study groups; 697.2 ± 91.44 SD versus 679.7 ± 75.40 SD treated with conventional healing caps, regarding age (Figure 15).

According to the voxel measurements after treatment with MED pulse electromagnetic healing caps (Figure 16) there was a significant difference *p* < 0.0001 between bone radiodensity before treatment 310.7 ± 53.26 SD and bone radiodensity after treatment 734 ± 61.96 SD.

Sub-analysis of the radiodensity in patients treated with pulse electromagnetic healing caps regarding age (Figure 17), showed a statistical difference *p* = 0.03 between study group 1 (30–45 years old) 745.9 ± 64.05 SD versus study group 2 (45–60 years old) 684.5 ± 74.23 SD. 

No statistical differences *p* = 0.10 were recorded between voxel value regarding bone radiodensity before treatment either with conventional caps 288.1 ± 47.16 SD or pulse electromagnetic healing caps; 310.7 ± 53.26 (Figure 18).

No statistical differences *p* = 0.12 were recorded between voxel value regarding bone radiodensity after treatment either with conventional caps 697.2 ± 91.44 SD or pulse electromagnetic healing caps 745.9 ± 64.05 SD (Figure 19).

According to the voxel measurements after treatment with conventional healing caps (Figure 20), there was a significant difference *p* < 0.0001 between bone radiodensity before treatment 312.0 ± 47.31 Standard Deviation (SD) and bone radiodensity 471.3 ± 60.99 SD after treatment with conventional healing caps of patients with orthodontic therapy in medical history. 

According to the voxel measurements after treatment with MED healing caps (Figure 21) there was a significant difference *p* < 0.0075 between bone radiodensity before implant placement 319.2 ± 51.44 Standard Deviation (SD) and bone radiodensity 514.4 ± 100. SD after implant placement with MED healing caps of patients with orthodontic therapy in medical history. 

## 5. Discussion

The use of the HU to measure tissue density has aided radiologists in the interpretation of images and diagnosis of disease. One of the earliest uses of the HU as a quantitative measurement came in the evaluation of solitary pulmonary nodules. More recently, HU measurements of bone on quantitative CTs and conventional CTs have helped to determine bone mineral density. Hounsfield unit measurements of bone have also been the object of recent study to estimate bone quality [10].

Lang et al. [11] reported preliminary radiographic densitometric studies of the interface region that might be considered an approximation of actual bone contact with implants. The most common findings in preliminary studies were a slightly higher radiodensity nearest the interface, which remained largely unchanged on follow-up radiographs. In contrast, one out of every ten osseointegrated implants demonstrated a pronounced radiodensity immediately surrounding the implant, which may or may not have increased in time. Failed implants often exhibited an area of relative radiolucency at the interface, but there was no evidence to demonstrate that failure could be predicted with this technique.

In agreement with established evidence, the authors noted that the mandible generally had a higher cancellous radiodensity compared with the maxilla, and that certain combinations of bone quantity and quality were often seen together. Sixty of these observations have since been corroborated independently [12]. It appears, for example, that Type D or E bone (advanced resorption) in the anterior part of the mandible may tend to have a high quality rating, often Type 1 or 2, since it is usually associated with dense basal bone. In comparison, Type A or B bone in the anterior part of the maxilla often has a Type 3 or 4 quality since alveolar bone in the maxilla usually has a very thin buccal cortex and a relatively low cancellous density. The significance of poor bone quantity for implant treatment planning is perhaps obvious in that it can restrict the surface area available for osseointegration [9].

Further studies [13] regarding pulsed electromagnetic treatment showed enhanced bone healing increased neovascularization and cell ingrowth within necrotic bone. Authors propose that the electromagnetic use, noninvasive physical therapy, may be used to enhance fracture healing and stimulated neovascularization in the necrotic grafted bone at the early stages of transplantation. This circulation improvement may lead to BMSC recruitment and nutrient supplement [13].

Only long-term randomized controlled clinical trials can give a definitive answer on the best way to treat lost implants. The long-term success of an implant depends on the regular maintenance schedule. During the maintenance phase, the perimeter tissue should be evaluated for inflammation. X-rays will show the bone issues around the implants. To date, no methodology has been established as a gold standard for the most correct use of gingival healing caps [14].

Controlled trials combined with comparative randomized clinical trials of healing caps pulsating electromagnetic waves are limited in number and have short follow-up periods and small sample sizes, thus presenting a high risk of bias [15,16]. Pulsed electromagnetic fields (PEMF) have been used for several years to supplement bone healing, because they can also accelerate intramedullary angiogenesis and improve the load to failure and stiffness of the bone. Although healing rates have been reported in up to 87% of delayed unions and non-unions, the efficacy of the method varied significantly while patient or fracture related variables could not be clearly associated with a successful outcome [16]. It is still doubtful which of the therapeutic strategies are the most effective for the treatment of peri-implant lesions depending on their morphology, extent, and severity. However, this does not suggest that the treatment modalities currently applied may not provide beneficial results in clinical practice [17].

## 6. Conclusions

(1)No statistical differences were observed regarding bone radiodensity after treatment with conventional healing caps vs. pulse electromagnetic stimulation caps, however, there was a difference regarding MED stimulation depending on the age of the patient, with younger patients experiencing a slightly increased degree of bone radiodensity.(2)The most important result of our study was a slightly higher radiodensity nearest the interface of dental implants, with the overall level of cancellous tissue density especially including variation in the level of trabecular bone mineralization.(3)A significant limitation of this study is the lack of a large dataset, which would be needed to offer further insight regarding pulsed electromagnetic waves and their influence on the bone. Additional research, using randomized controlled trials, should be conducted to ascertain its effectiveness compared with other treatment modalities.

## Figures and Tables

**Figure 1 jcm-09-03983-f001:**
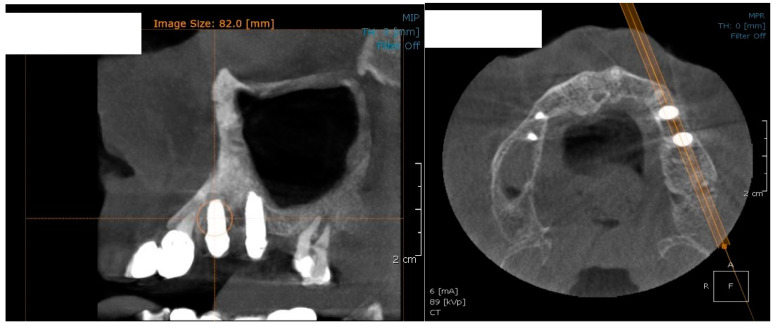
The field of view was a cylinder of 82 mm in diameter and height. The **left** panel is a 3D section CBCTCone Beam Computed Tomography image, whereas the **right** panel represents an oblique-section of the same image. F, frontal view, A, Axial View, R, right side view.

**Figure 2 jcm-09-03983-f002:**
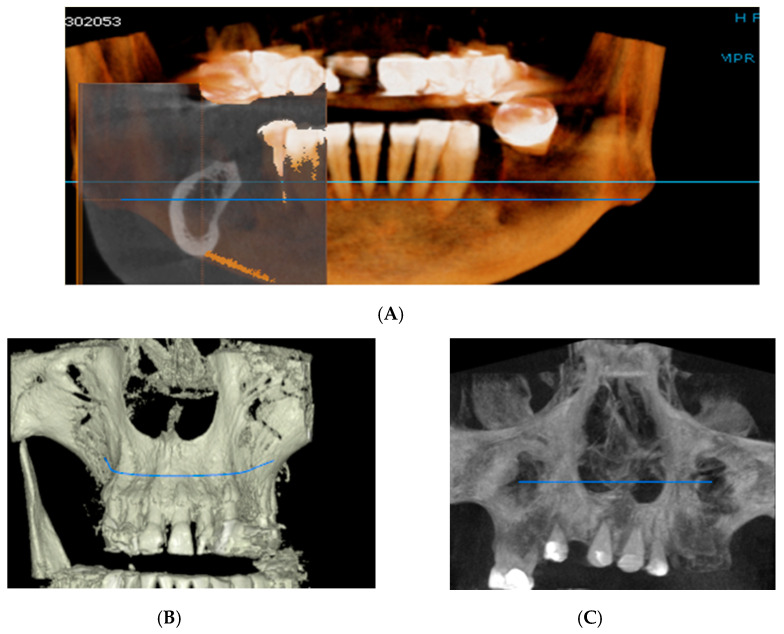
Rendering images showing field of view (FOV) images used in the study. (**A**) Frontal view, (**B**) Frontal Bone view, (**C**) X-ray view.

**Figure 3 jcm-09-03983-f003:**
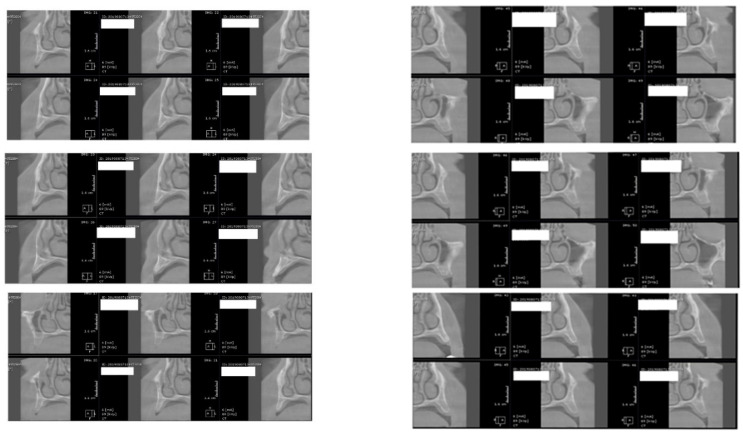
CBCT images used in the study, showing sagittal sections of bone.

**Figure 4 jcm-09-03983-f004:**
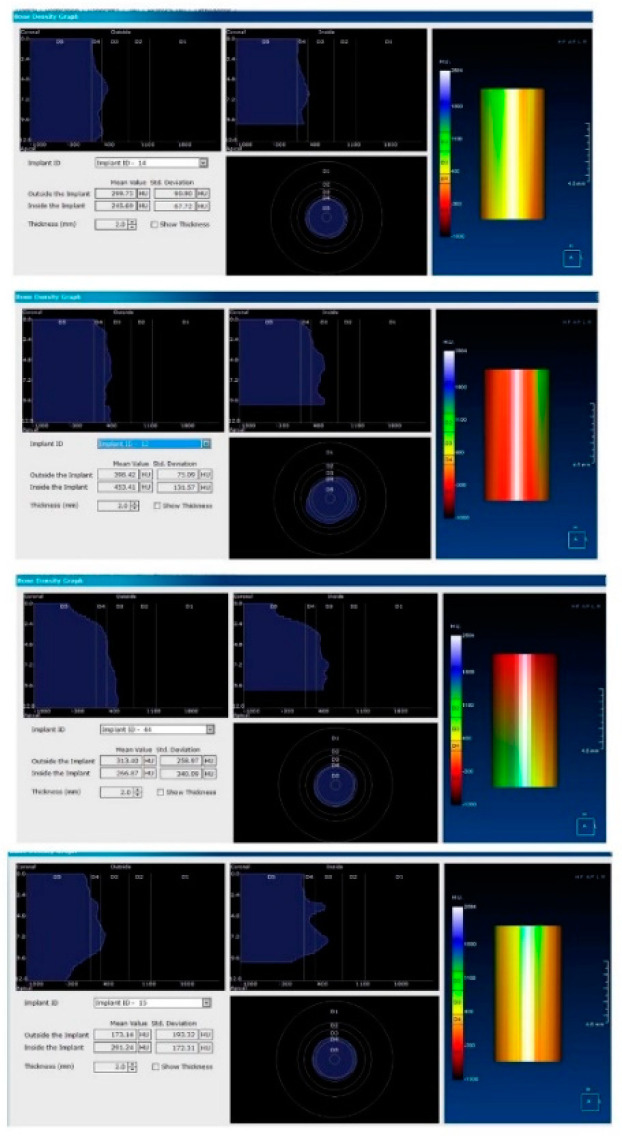
CBCT images used in the study showing the density of bone.

**Figure 5 jcm-09-03983-f005:**
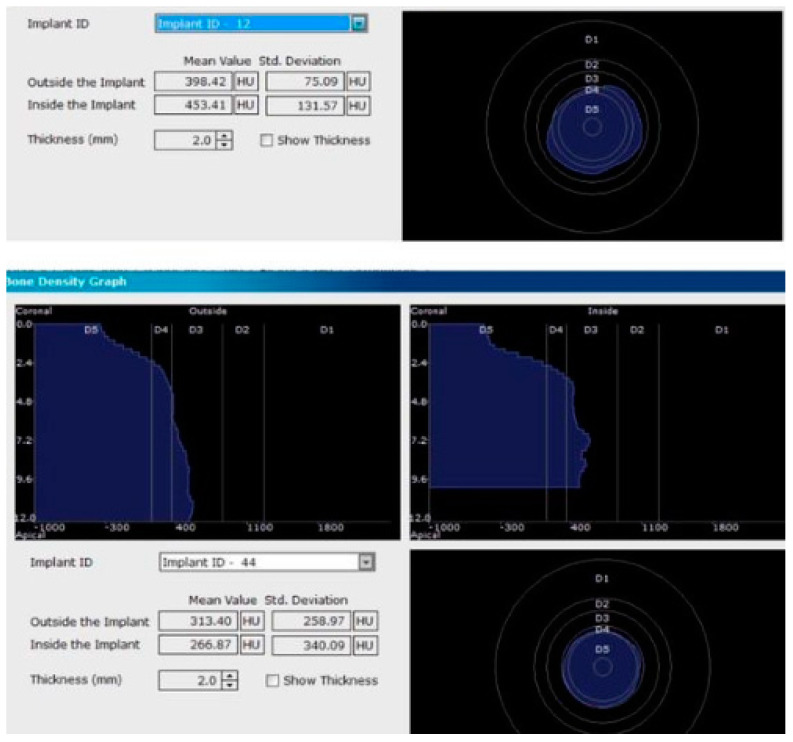
Image showing the bone density measured in Hounsfield units (HU) as well as the standard deviations of the volume.

**Figure 6 jcm-09-03983-f006:**
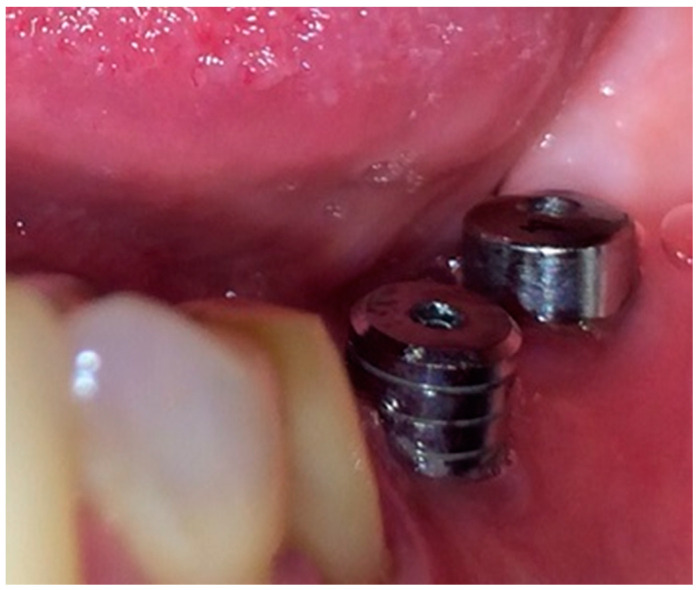
Conventional healing caps in position.

**Figure 7 jcm-09-03983-f007:**
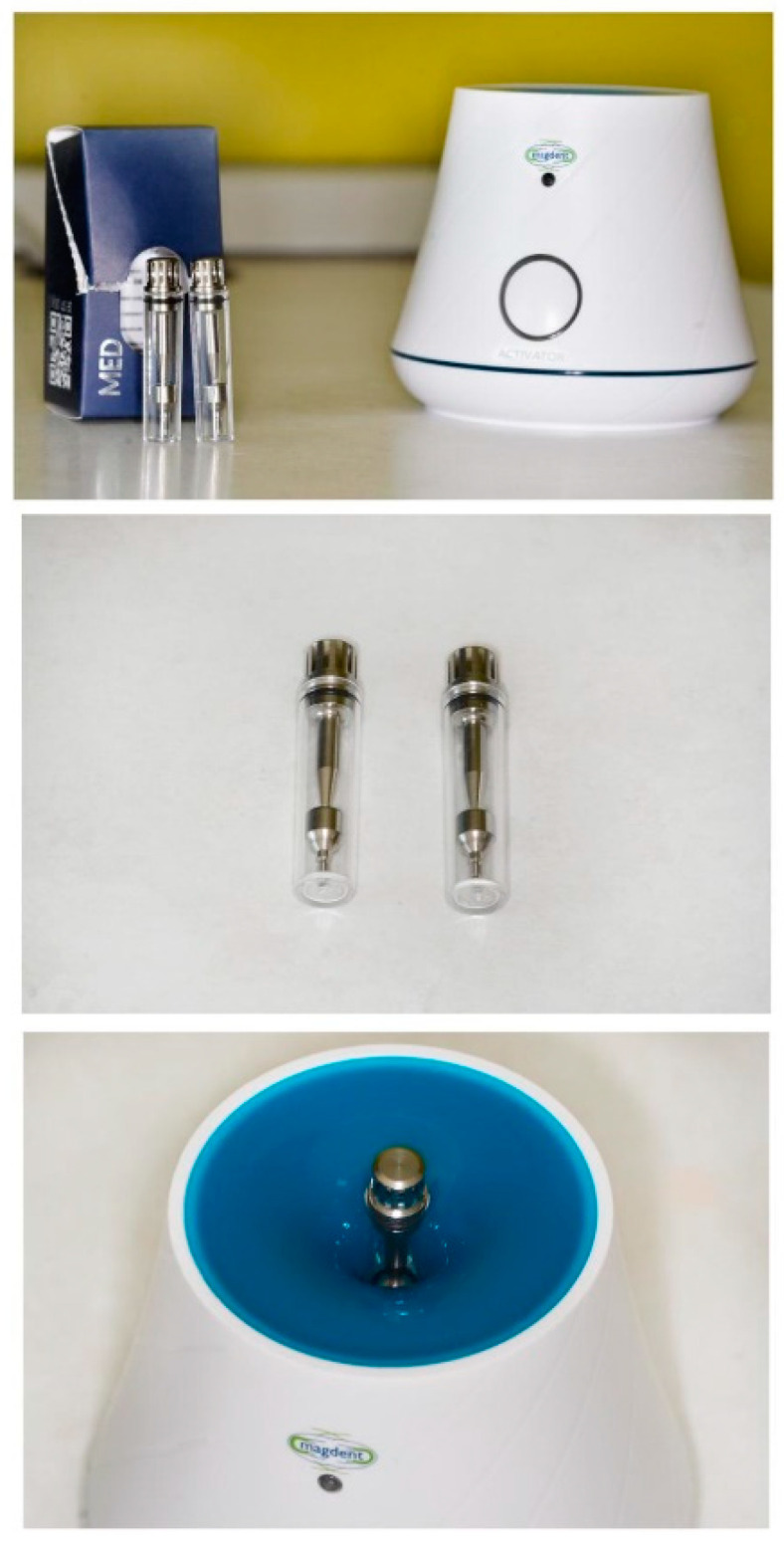
Magdent™ pulse electromagnetic healing caps.

**Figure 8 jcm-09-03983-f008:**
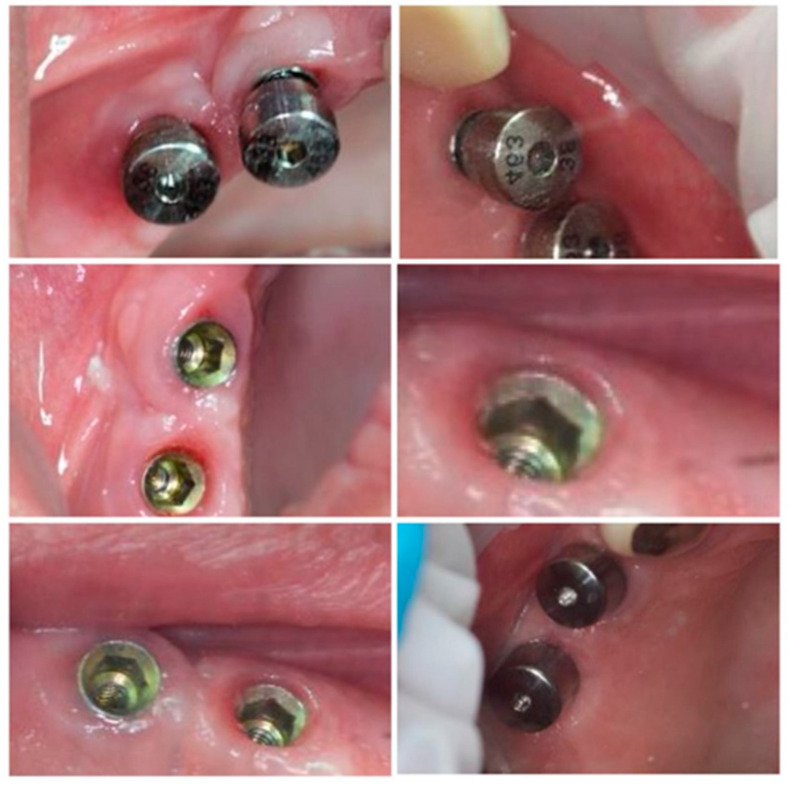
Magdent™ pulse electromagnetic healing caps in position.

**Figure 9 jcm-09-03983-f009:**
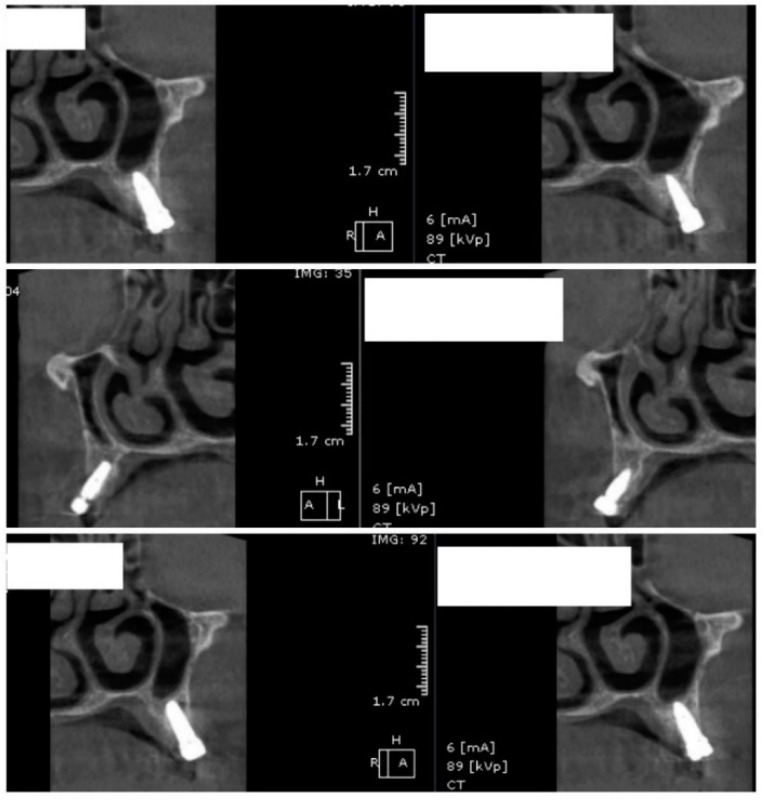
Random CBCT images with conventional healing caps in place.

**Figure 10 jcm-09-03983-f010:**
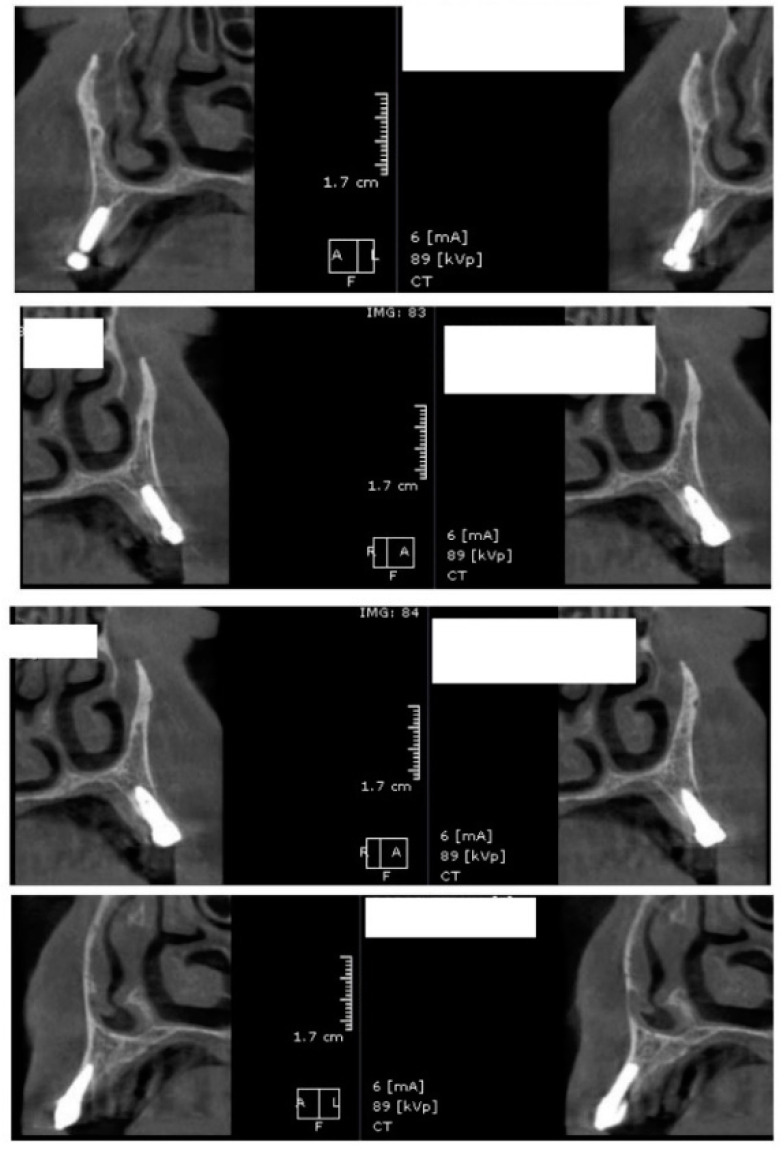
Random CBCT sections with pulse electromagnetic healing caps in position.

**Figure 11 jcm-09-03983-f011:**
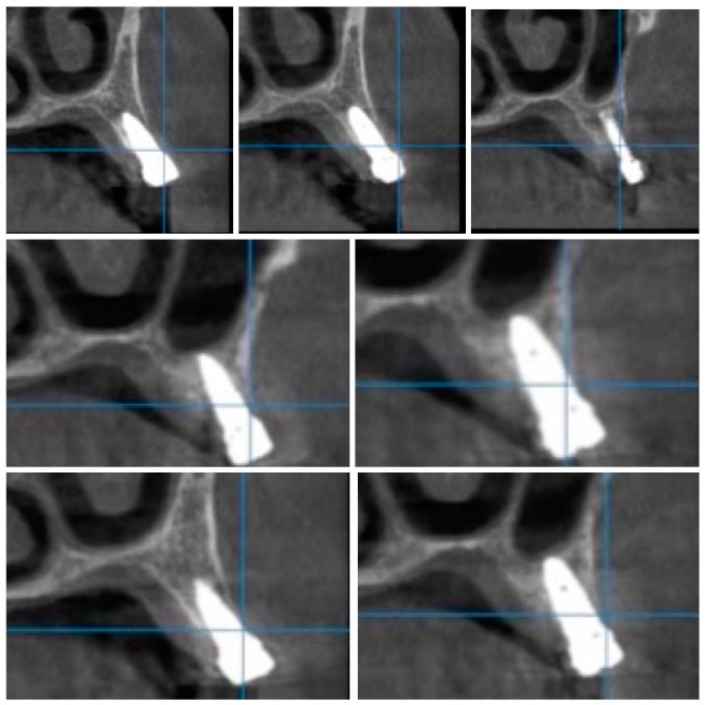
Random CBCT images with pulse electromagnetic healing caps in position.

**Figure 12 jcm-09-03983-f012:**
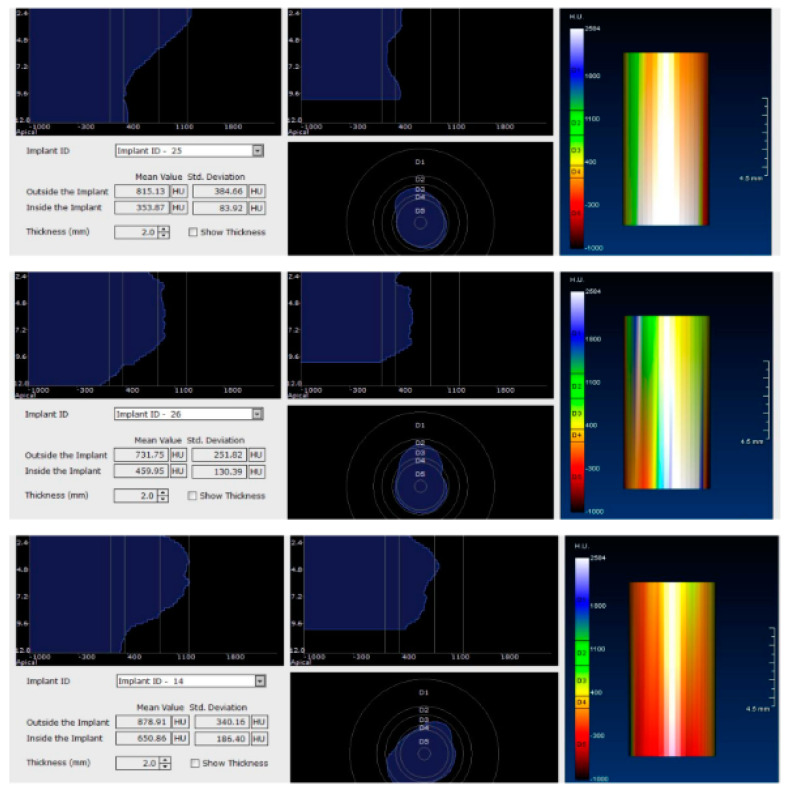
Bone density around dental implants at 60 days following surgery with conventional healing caps.

**Figure 13 jcm-09-03983-f013:**
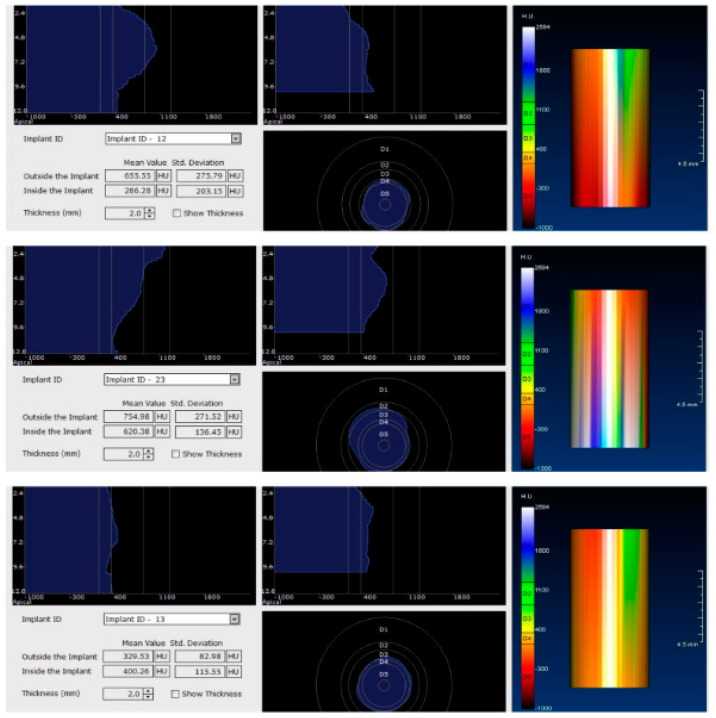
Bone density around dental implants at 60 following surgery with MED™ electromagnetic healing caps.

**Figure 14 jcm-09-03983-f014:**
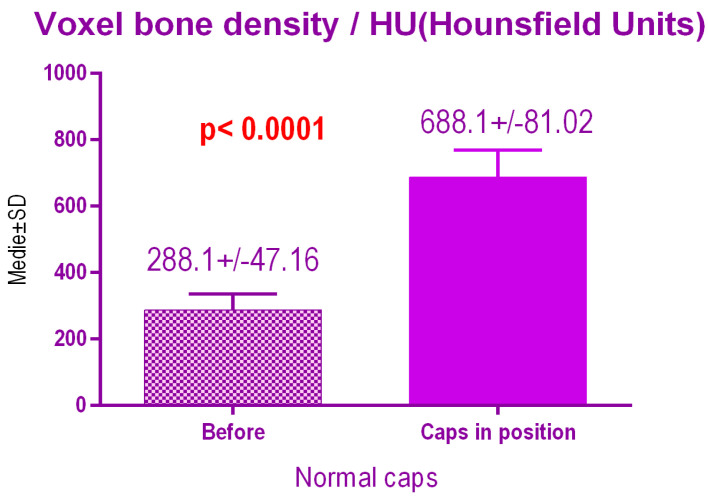
Bone radiodensity after conventional healing caps.

**Figure 15 jcm-09-03983-f015:**
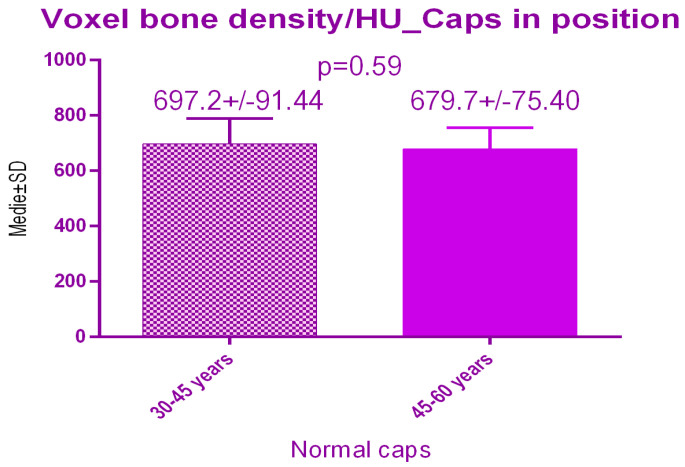
Bone radiodensity comparison by age after conventional healing caps.

**Figure 16 jcm-09-03983-f016:**
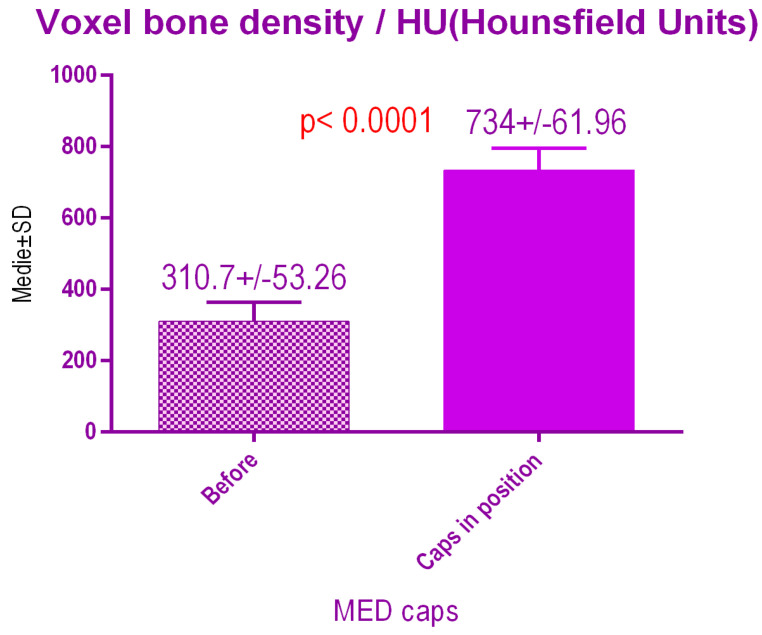
Bone radiodensity after pulse electromagnetic healing caps.

**Figure 17 jcm-09-03983-f017:**
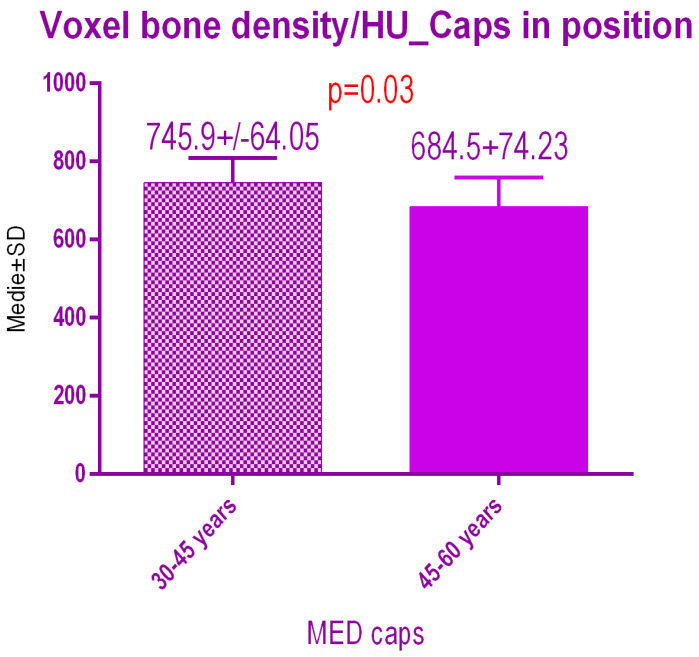
Bone radiodensity comparison by age after pulse electromagnetic healing caps.

**Figure 18 jcm-09-03983-f018:**
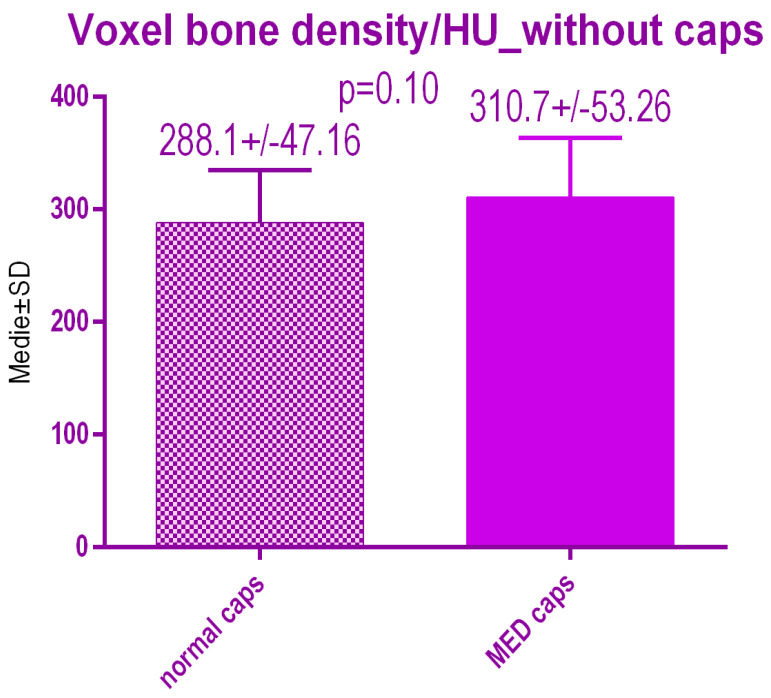
Bone radiodensity in the study groups before treatment.

**Figure 19 jcm-09-03983-f019:**
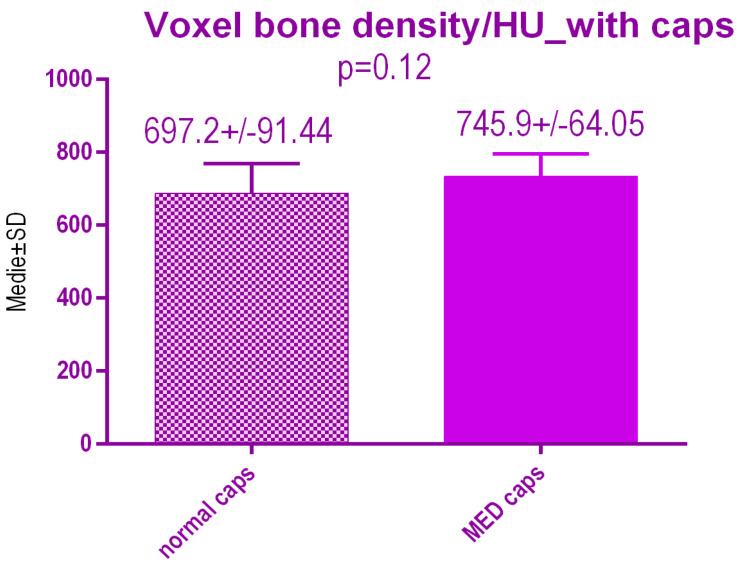
Bone radiodensity in the study groups after treatment.

**Figure 20 jcm-09-03983-f020:**
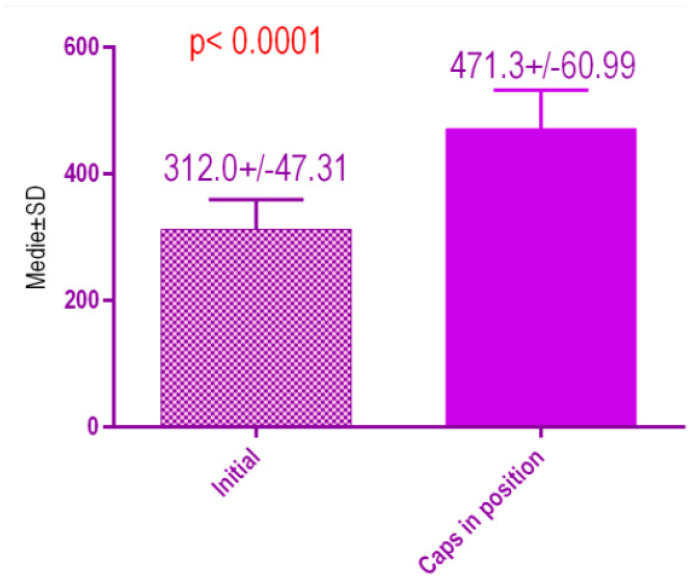
Bone radiodensity in the study groups, patients with orthodontic therapy in medical history, before vs. after treatment with conventional healing caps.

**Figure 21 jcm-09-03983-f021:**
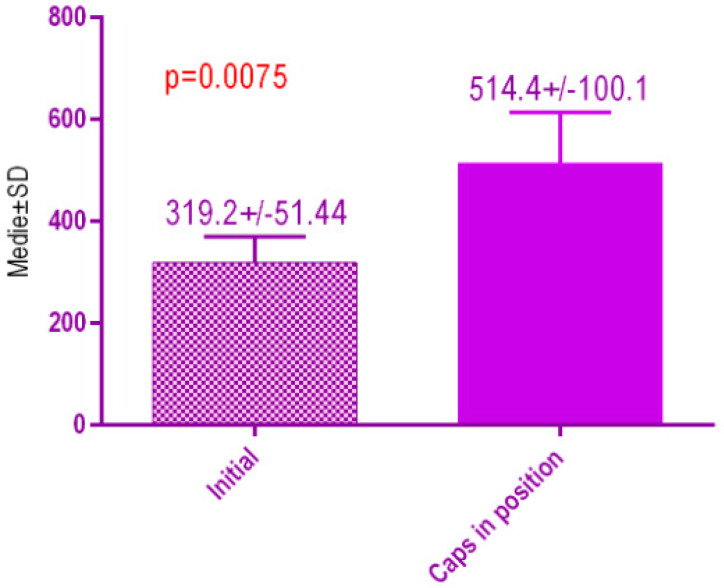
Bone radiodensity in the study groups, patients with orthodontic therapy in medical history, before vs. after treatment with MED healing caps.

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
