# Peer review of "Clinical Studies Regarding Electromagnetic Stimulation in Proximity of Dental Implants on Patients with/without Orthodontic Treatment"

_jcm, 2020, doi:10.3390/jcm9123983_

Round 1
Reviewer 1 Report
In the introduction at no time do they speak of the influence of orthodontic treatments on bone quality. Neither in the material and method do they mention patients with orthodontic treatments. Therefore the title does not have much to do with the subject of the study.
The bibliographic references do not correspond to the numbering of the text (Example reference nº4 "experimentation with animals" which should be nº5).
On page 3 the references to the figures is not correct since it would be Figs 3 and 4 but not 5 that appears later (on page 5).
In the description of the surgical stage, figure 5 appears again (when it is figure 6).
It would be interesting to know if there is any type of study or article regarding electromagnetic waves and their influence on bone, even if it was not at the level of the oral cavity.
In the results, they do not analyze whether the Ostell values ​​have any influence on the final bone quality, regardless of the type of treatment performed on the patient.
In the discussion, much emphasis is placed on tobacco as one of the main causes of implant failure, but other causes such as the surgical act, the three-dimensional position of the implant, and occlusion are hardly developed. Therefore in my opinion that part can be improved.
Author Response
Please see the attachment.
Dear Reviewer,
We would like to thank you for taking the time to assess your manuscript. We have read with great interest your remarks regarding our manuscript entitled ” Clinical studies regarding electromagnetic stimulation in proximity of dental implants on patients with/ without orthodontic treatment “ . We would like to inform you that following your suggestions we have made the following modifications to the manuscript and we have addressed all the concerns you raised. All the changes made to the manuscript are highlighted with the “ Track Changes” tool of the Microsoft Word software in red color.
- In the introduction at no time do they speak of the influence of orthodontic treatments on bone quality. Neither in the material and method do they mention patients with orthodontic treatments. Therefore the title does not have much to do with the subject of the study.
We thank the reviewer for pointing this out. We have revised the Introduction part and we added information regarding the influence of orthodontic treatment in bone quality (please see page 1), also in the material and method part we added data regarding details of patients which underwent orthodontic treatment prior the moment of our study. (please see page 5)
- The bibliographic references do not correspond to the numbering of the text (Example reference nº4 "experimentation with animals" which should be nº5).
We apologize for our error, we corrected reference number 5, instead of reference number 4 (page 1, line 33), thank you for pointing this out.
- On page 3 the references to the figures is not correct since it would be Figs 3 and 4 but not 5 that appears later (on page 5).
In the description of the surgical stage, figure 5 appears again (when it is figure 6).
We agree with the reviewer and following reviewer suggestion we have fixed the error(page 5, line 134).
- It would be interesting to know if there is any type of study or article regarding electromagnetic waves and their influence on bone, even if it was not at the level of the oral cavity.
We appreciate the reviewer’s insightful suggestion and agree that it would be interesting to evaluate electromagnetic waves and their influence on bone healing process in the oral cavity. We have found few studies in the literature, however we consider they are not reliable since they were conducted by the manufacturer of this device: Effect of pulsed electromagnetic field on healing of mandibular fracture: a preliminary clinical study. Delrahim A, Hassanein HR, Dahaba M (2011) Journal of Oral and Maxillofacial Surgery : Official Journal of the American Association of Oral and Maxillofacial Surgeons [01 Feb 2011, 69(6):1708-1717]; Miniaturized electromagnetic device abutment improves stability of the dental implants S. Barak, S. Matalon, B. Zavan, A. Piattelli .
Regarding electromagnetic waves and their influence on bone healing process a lot of researchers analised this method of treatment in other medical specialities, expecially ortopaedics: A novel single pulsed electromagnetic field stimulates osteogenesis of bone marrow mesenchymal stem cells and bone repair.Fu YC, Lin CC, Chang JK, Chen CH, Tai IC, Wang GJ, Ho ML.PLoS One. 2014 Mar 14;9(3):e91581. doi: 10.1371/journal.pone.0091581. eCollection 2014; Stimulation of bone formation and fracture healing with pulsed electromagnetic fields: biologic responses and clinical implications.Chalidis B, Sachinis N, Assiotis A, Maccauro G.Int J Immunopathol Pharmacol. 2011 Jan-Mar;24(1 Suppl 2):17-20. doi: 10.1177/03946320110241S204.
We also added these informations in the Discussion section(please see page 15)
- In the results, they do not analyze whether the Ostell values ​​have any influence on the final bone quality, regardless of the type of treatment performed on the patient
We agree with the reviewer that further elaborating on this point using new data would be helpful. However, we believe that expanding our dataset would not significantly support our argument because all the implants were osteointegrated at the moment of our analisys. We did not want to evaluate the implant stability, we have analised only the changes that may occur at bone level radiodensity before the treatment and after treatment around dental implants.
- In the discussion, much emphasis is placed on tobacco as one of the main causes of implant failure, but other causes such as the surgical act, the three-dimensional position of the implant, and occlusion are hardly developed. Therefore in my opinion that part can be improved
We have revised the text to address your concerns and hope that it is now clearer. Please see page 15 of the revised manuscript.
We would like to thank the reviewer again for taking the time to review our manuscript.

Reviewer 2 Report
Dear authors, this study is a very interesting study. However, I have found some aspects to be considered:- Conclusions are somewhat like discussion. Conclusions should be based on the scientifically-significant findings only.
- Patients with orthodontic treatment history was tested. However, statistical analysis was not applied at all for these tests. Why were these measurements made?
- Bone densities are quite different between maxilla and mandible. Furthermore, bone densities are different among areas within an arch. Why were jaws and areas within a jaw not considered?
- The method error or reproducibility of the measurements should be tested.
- In the CBCT, HU is not accurate relative to the conventional fan-beam CTs. Therefore, the term HU is rarely used in CBCT studies. Finding different terminology is recommended.
- After all, the pulse electromagnetic healing caps does not seem to provide any benefit. To conclude like this, use of more larger sample is desirable.
Author Response
Please see the attachment.
Dear Reviewer,
We would like to thank you for taking the time to assess your manuscript. We have read with great interest your remarks regarding our manuscript entitled ” Clinical studies regarding electromagnetic stimulation in proximity of dental implants on patients with/ without orthodontic treatment “ . We would like to inform you that following your suggestions we have made the following modifications to the manuscript and we have addressed all the concerns you raised. All the changes made to the manuscript are highlighted with the “ Track Changes” tool of the Microsoft Word software in red color.
- Conclusions are somewhat like discussion. Conclusions should be based on the scientifically-significant findings only.
We thank the reviewer for pointing this out. We have revised the Conclusions part and we added information as suggested (please see page 16 )
- Patients with orthodontic treatment history was tested. However, statistical analysis was not applied at all for these tests. Why were these measurements made?
We have revised the text to address your concerns and hope that it is now clearer. Since some patients enrolled in our study underwent orthodontic treatment for space opening on the arch, prior to insertion of dental implants, we wanted to analise if some of these patients may present bone quality modification after orthodontic treatment as sugested by the following researchers: Changes in the alveolar bone thickness of maxillary incisors after orthodontic treatment involving extractions - A systematic review and meta-analysis. Domingo-Clérigues M, Montiel-Company JM, Almerich-Silla JM, García-Sanz V, Paredes-Gallardo V, Bellot-Arcís C.J Clin Exp Dent. 2019 Jan 1;11(1):e76-e84. doi:10.4317/jced.55434.eCollection 2019 Jan.PMID: 30697398; Assessment of the changes in alveolar bone quality after fixed orthodontic therapy: A trabecular structure analysis. Haghnegahdar A, Zarif Najafi H, Sabet M, Saki M.J Dent Res Dent Clin Dent Prospects. 2016 Fall;10(4):201-206. doi: 10.15171/joddd.2016.032. Epub 2016 Dec 21.
We agree and have updated informations. (please see page 5 )
- Bone densities are quite different between maxilla and mandible. Furthermore, bone densities are different among areas within an arch. Why were jaws and areas within a jaw not considered?
We agree with the reviewer that further elaborating on this point using new data would be helpful. However, we believe that expanding our dataset is neither feasible, given the costs involved, nor would significantly support our argument,because we did not wanted to perform a analisys at different levels of the jaw on the same patient, we measured the radiodensity of the bone prior to the insertion of the implant and after osteointegration of the implants eighter the implants were loaded with conventional healing caps or electromagnetic pulse caps.
- The method error or reproducibility of the measurements should be tested
We appreciate the reviewer’s insightful remark and agree that artifacts and metal effect of the dental implants may produce errors,however calibration at the same parameters of the CBCT unit prior to, and after examination was made thus reducing the risk of large errors during CBCT exposure. Nevertheless, we recognize this limitation should be mentioned in the paper, so we added the following sentence: A big limitation of this study is the lack of a large dataset which would be needed to offer further insight regarding pulsed electromagnetic waves and their influence on the bone. Additional research, using randomized controlled trials, should be conducted to ascertain its effectiveness compared with other treatment modalities.(please see page 17)
- In the CBCT, HU is not accurate relative to the conventional fan-beam CTs. Therefore, the term HU is rarely used in CBCT studies. Finding different terminology is recommended.
We thank the reviewer for pointing this out , however we based our study on the following studies found in the literature: CBCT-based bone quality assessment: are Hounsfield units applicable? Pauwels R, Jacobs R, Singer SR, Mupparapu M. Dentomaxillofac Radiol. 2015;44(1):20140238. doi: 10.1259/dmfr.20140238.; Hounsfield Units on Lumbar Computed Tomography for Predicting Regional Bone Mineral Density. Kim KJ, Kim DH, Lee JI, Choi BK, Han IH, Nam KH. Open Med (Wars). 2019 Jul 19;14:545-551. doi: 10.1515/med-2019-0061. eCollection 2019.; Hounsfield Unit Tami D. DenOtter , Johanna Schubert In: StatPearls [Internet]. Treasure Island (FL): StatPearls Publishing; 2020 Jan. 2020 May 11, PMID: 31613501. As the measurement unit of radiodensity is expressed in Hounsfield Units (HU) in metrology SI, we found more suitable to use this term during our study. We also added informations regarding this measurement unit in the discussion section(please see page 15)
- After all, the pulse electromagnetic healing caps does not seem to provide any benefit. To conclude like this, use of more larger sample is desirable
We agree with the reviewer that further research on this field using new data would be helpful.We agree that a limitation of our study is the low sample size of the study population and further research in this field needs to be conducted so we added the following sentence: A big limitation of this study is the lack of a large dataset which would be needed to offer further insight regarding pulsed electromagnetic waves and their influence on the bone. Additional research, using randomized controlled trials, should be conducted to ascertain its effectiveness compared with other treatment modalities. (please see page 17)
We would like to thank the reviewer again for taking the time to review our manuscript.

Round 2
Reviewer 1 Report
Thank you very much for the corrections. In my opinion the article is already well structured and justified to be published. I hope you take my views into account for future publications.
Author Response
Thank you very much for the corrections. In my opinion the article is already well structured and justified to be published. I hope you take my views into account for future publications.
Once again, we thank you for the time you put in reviewing our paper and look forward to meeting your expectations. Since your inputs have been precious we will take them into consideration in our future research projects.
The authors’
Reviewer 2 Report
Dear authors,
Although the history of orthodontic treatment was included in the title, the results did not show considerable attention to the effect of the history of orthodontic treatment, and this was not discussed at all. Statistical comparison of the improvement of bone density between conventional group and MED group is recommended.
Thank you for your effort to improve the manuscript.
Author Response
Although the history of orthodontic treatment was included in the title, the results did not show considerable attention to the effect of the history of orthodontic treatment, and this was not discussed at all. Statistical comparison of the improvement of bone density between conventional group and MED group is recommended.
Once again, we thank you for the time you put in reviewing our paper and look forward to meeting your expectations. Since your inputs have been precious to us we have revised the text and made the following modifications:
- In the results section we added data regarding bone modifications that might have occured after orthodontic treatment prior to our study.(please see pages 14-15, lines 248-260)
- In the discussion section we added informations regarding bone modifications during orthodontic treatment(please see page 16, lines 303-308).
- In the conclusion section we also added informations regarding orthodontic treatment and possible bone modifications under orthodontic forces.(please see page 17, lines 349-351.)
This manuscript is a resubmission of an earlier submission. The following is a list of the peer review reports and author responses from that submission.